# Specific stereochemistry of OP-1074 disrupts estrogen receptor alpha helix 12 and confers pure antiestrogenic activity

S.W. Fanning[1], L. Hodges-Gallagher[2], D.C. Myles[2], R. Sun[2], C.E. Fowler[1], I.N. Plant[3], B.D. Green[1], C.L. Harmon[2], G.L. Greene ⓘ [1] & P.J. Kushner[2]

Complex tissue-specific and cell-specific signaling by the estrogen receptor (ER) frequently leads to the development of resistance to endocrine therapy for breast cancer. Pure ER antagonists, which completely lack tissue-specific agonist activity, hold promise for preventing and treating endocrine resistance, however an absence of structural information hinders the development of novel candidates. Here we synthesize a small panel of benzopyrans with variable side chains to identify pure antiestrogens in a uterotrophic assay. We identify OP-1074 as a pure antiestrogen and a selective ER degrader (PA-SERD) that is efficacious in shrinking tumors in a tamoxifen-resistant xenograft model. Biochemical and crystal structure analyses reveal a structure activity relationship implicating the importance of a stereospecific methyl on the pyrrolidine side chain of OP-1074, particularly on helix 12.

[1] Ben May Department for Cancer Research, University of Chicago, Chicago, IL 60637, USA. [2] Olema Pharmaceuticals, San Francisco, CA 94107, USA. [3] Department of Systems Biology, Harvard Medical School, Boston, MA 02115, USA. These authors contributed equally: S.W. Fanning, L. Hodges-Gallagher. These authors jointly supervised this work: G.L. Greene, P.J. Kushner. Correspondence and requests for materials should be addressed to S.W.F. (email: sfanning@uchicago.edu) or to L.H.-G. (email: leslie@olemapharma.com)

Endocrine therapy remains a cornerstone of treating estrogen receptor alpha (ERα)-positive breast tumors by blocking the agonistic effects of estrogen, either by directly antagonizing receptor binding or by blocking its production with aromatase inhibitors. A serious limitation of these drugs is the development of resistance that frequently occurs after prolonged use. Fulvestrant is a pure antiestrogen and selective ER degrader (PA-SERD) that is approved to treat hormone responsive tumors in postmenopausal women whose disease has progressed following treatment with tamoxifen and aromatase inhibitors. Fulvestrant also recently demonstrated benefit as a first-line therapy in the phase III FALCON trial, where it significantly increased progression-free survival over the aromatase inhibitor anastrazole[1]. Unfortunately, fulvestrant has poor pharmacokinetic properties, requires intramuscular delivery, and does not fully saturate the receptor even at the higher 500 mg dose[2]. Thus, fulvestrant is unlikely to achieve its full therapeutic potential, particularly in premenopausal women, where even a 750 mg dose was inferior to tamoxifen[3].

Selective estrogen receptor modulators (SERMs), such as tamoxifen, exhibit tissue-specific agonist activity in the bone and uterine endometrium but antagonize ER signaling in the breast. Importantly, this partial agonism is implicated in the switch from tamoxifen-responsive tumors to the development of resistance[4–6]. In addition, cells in which tamoxifen displays agonist activity are highly dependent on ER activation function-1 (AF-1) activity[7,8]. SERMs stimulate the transcription of estrogen responsive genes dependent on AF-1, and phosphorylation of AF-1 by growth factors further enhances agonist activity in a ligand-independent manner[9,10]. This crosstalk between ERα and growth factor signaling has been shown to play a role in the development of tamoxifen-resistance[11–13]. Enhanced activation of genes containing AP-1 sites is also associated with tamoxifen resistance in patients, and blocking AP-1 can reverse resistance in breast cancer models[14–17]. Tamoxifen, but not fulvestrant, can stimulate the transcription of hormone-responsive promoters at AP-1 sites via an ER/AP-1 complex in uterine endometrial cells, but not in breast cells, paralleling the tissue-specific agonism observed in vivo[18]. In contrast to the partial agonism observed with tamoxifen and other SERMs, fulvestrant is a pure antiestrogen: i.e., it lacks agonist activity in all hormone-sensitive tissues and completely antagonizes E2-stimulated signaling and proliferation[19,20]. It has been postulated that fulvestrant is effective as a second-line therapy because of its ability to completely block this promiscuous ER-mediated signaling characteristic of SERMs[21,22]. However, a revision of this model may be required in light of results obtained from a recent phase I study, in which the SERM endoxifen showed clinical benefit in treating patients whose tumors had progressed with prior fulvestrant treatment[23].

Ligand binding mediates distinct changes in receptor conformation in the ligand binding domain (LBD), leading to differential recruitment of coregulatory molecules, binding to specific response elements, and post-translational modifications, such as receptor degradation. Agonist binding to the ERα LBD stabilizes helix 12 (H12) docking between H3 and H11 to expose the activating function 2 (AF-2) cleft for coregulator binding to a conserved LXXLL motif (Supplemental Fig. 1A)[24]. Antagonists bind in the hormone-binding pocket and extend their side-chains outwards, preventing the agonist conformation of H12, which docks in the AF-2 cleft with its LXXML motif and blocks coactivator binding, promotes corepressor binding complexes, and affects downstream transcription (Supplementary Fig. 1B)[25]. SERDs such as fulvestrant, possess bulkier or extended side-chains that more fully disrupt H12 leading to proteosomal degradation[26,27], although pure antagonism and receptor degradation can also occur in the absence of a prototypical side chain

by dislocating helix 11[28]. The orientation of H12 also plays an important role in corepressor binding, with fulvestrant-bound ER adopting a conformation that allows for more efficient recruitment[19].

We recently described crystal structures that provided insight into how activating mutations in the LBD confer constitutive activity often associated with resistance[29]. In particular, D538 and Y537 mutants stabilize H12 in an active conformation that enables coactivator recruitment in the absence of E2, consequently resulting in reduced ligand affinity, as well as reduced potency and efficacy by antiestrogens. These studies highlight the important nature of H12 as an essential molecular determinant for antiestrogen action, and a more comprehensive understanding of this interaction will aid in designing the next generation of inhibitors capable of treating clinical resistance.

Here we use a series of benzopyran-based ligands to reveal how variations in side chain composition affects agonist/antagonist activities, using methods that discriminate between SERMs and pure antiestrogens in vitro and in vivo. Subsequent biophysical and structural analyses reveal how subtle chemical changes to the methylpyrrolidine side chain alter the conformation of antiestrogen-bound ER and mobility of H12.

## Results

**Alkaline phosphatase activity predicts OP-1074 is a pure antiestrogen.** OP-1074 (R-3-(4-hydroxyphenyl)-4-methyl-2-(4-2-3-methylpyrrolidin-1-yl)ethoxy)phenyl)-2Hchromen-7-ol) and analogs were synthesized for the purpose of investigating the structural and biological effects of methyl substitutions on the pyrrolidine side chain on a benzopyran scaffold. Our purpose in designing these analogs was to explore the stereochemical space of the antiestrogen side chain shown to be involved in disrupting the active LBD conformation and occlude coactivator binding by repositioning H12[25,26,30]. The structure of OP-1074 and analogs and a summary of results in the in vitro biological assays are shown in Fig. 1 and Table 1, respectively. We also confirmed that OP-1074 lacked activity on other steroid receptors and had specific antiestrogenic activity for both ERα and ERβ, where it inhibited 17β-estradiol (E2)-stimulated transcriptional activity with IC$_{50}$'s of 1.6 and 3.2 nM, respectively (Supplementary Fig. 2).

Our primary screen to distinguish pure antiestrogens from SERMs was induction of alkaline phosphatase (AP) activity in

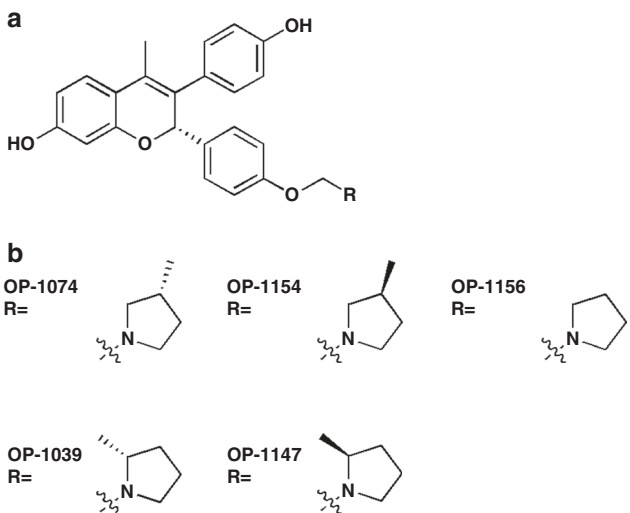

**Fig. 1** Benzopyran-based estrogen receptor alpha inhibitors. Structure of benzopyrene scaffold (**a**) and side chain (**b**) of OP-1074 and related analogs used in this study

**Table 1 Summary of data for in vitro assays**

| | Induction of AP activity[a] (% E2) | Relative ERα protein levels[b] (% vehicle) | IC$_{50}$ proliferation[c] (nM) | IC$_{50}$ transcription[d] (nM) |
|---|---|---|---|---|
| endoxifen | 73% ± 10 ($n = 8$) | 139% ± 34 ($n = 20$) | 13 ± 2.2 ($n = 5$) | 6.3 ± 1.3 ($n = 5$) |
| fulvestrant | 0.05% ± 2.7 ($n = 58$) | 28% ± 12 ($n = 40$) | 2.5 ± 0.3 ($n = 45$) | 2.0 ± 0.2 ($n = 41$) |
| OP-1074 | 10% ± 4.4 ($n = 29$) | 49% ± 13 ($n = 15$) | 7.2 ± 0.9 ($n = 26$) | 1.8 ± 0.3 ($n = 17$) |
| OP-1154 | 40% ± 6.5 ($n = 3$) | 123% ± 22 ($n = 4$) | 6.0 ± 1.1 ($n = 5$) | 5.6 ± 2.1 ($n = 4$) |
| OP-1039 | 49% ± 18 ($n = 8$) | 119% ± 13 ($n = 6$) | 6.4 ± 1.0 ($n = 10$) | 0.5 ± 0.1 ($n = 9$) |
| OP-1047 | 136% ± 15 ($n = 4$) | 319% ± 100 ($n = 5$) | 5.5 ± 1.4 ($n = 3$) | 1.3 ± 0.4 ($n = 3$) |
| OP-1156 | 40% ± 7.8 ($n = 3$) | 152% ± 31 ($n = 4$) | 5.2 ± 1.3 ($n = 5$) | 5.9 ± 2.1 ($n = 4$) |

Data are means obtained from multiple independent experiments ($n$), ±s.d.
[a]Maximum AP activity observed relative to positive control, 500 pM E2
[b]ERα protein levels relative to vehicle control obtained by immunoblotting
[c]Inhibition of proliferation of MCF-7 cells stimulated with 100 pM E2 after 5–7 days
[d]Inhibition of an estrogen responsive reporter gene, ERE-tk109-Luc, transiently transfected into MCF-7 cells and stimulated with 100 pM E2 for 24 h

Ishikawa uterine cells. Previous studies have demonstrated that this in vitro assay correlates with an ER-mediated increase in uterine wet weight in vivo and allows ligands to be quickly screened using considerably fewer resources[31–33]. The AP promoter contains two AP-1 sites and two putative estrogen response elements (EREs), and enzymatic induction of AP is a prime example of AF-1-dependent gene behavior induced by agonists and SERMs, e.g., tamoxifen, but not by the pure antiestrogen fulvestrant[34,35]. The reference SERM used in our screen was endoxifen (N-desmethyl-4-hydroxytamoxifen), the most abundant bioactive metabolite of tamoxifen[36]. In the agonist mode, that is in the absence of E2, endoxifen maximally stimulated AP activity 82% relative to E2 control, whereas our reference pure antiestrogen, fulvestrant, lacked any AP activity over all doses tested (Fig. 2a). In the antagonist mode of the AP assay, performed in the presence of 500 pM E2, endoxifen was a poor inhibitor of E2-induced activity, which still exhibited 58% activity even with 1 μM drug (Fig. 2b). In contrast, fulvestrant completely inhibited E2-stimulated AP activity in a dose responsive manner. These in vitro data correlate with uterine wet weights previously reported for ovariectomized rats treated for 3 weeks with tamoxifen and fulvestrant[37].

Of the panel of analogs in the current study, OP-1074 exhibited the least amount of AP activity in the agonist mode, with maximum activity of only 8% at 1 nM (Fig. 2a). In the antagonist mode OP-1074 completely antagonized E2, but was less potent than fulvestrant in inhibiting E2-stimulated AP activity, with an IC$_{50}$ of 20 nM, vs. 3 nM for fulvestrant (Fig. 2b). The unsubstituted pyrrolidine, OP-1156, induced AP activity 40%, in line with the increased uterine wet weight in ovariectomized mice previously reported for this compound[38]. In contrast to the antagonist activity observed for 3R-substituted pyrrolidine of OP-1074, the 3S-substituted analog, OP-1154, induced AP activity 39% and did not completely antagonize E2-stimulated AP activity. When comparing the AP activity relative to the position of methyl substitution on the pyrrolidine ring, the 3- position was less agonistic than the 2-position, and the R-orientation was less agonistic than the S- orientation (Fig. 2c).

To confirm that OP-1074 lacks agonist activity in the uterus, we tested it in ovariectomized BALB/c mice in a uterotrophic assay. Animals were treated with 100 mg/kg OP-1074 by oral gavage for 3 days in the presence or absence of E2, and changes in uterine wet weight measured. In contrast to E2 and tamoxifen, OP-1074 did not increase uterine wet weight (Fig. 2d). OP-1074 was not significantly different from vehicle control or fulvestrant in the agonist mode, and was also not significantly different from fulvestrant when co-treated with E2 in the antagonist mode. Thus, OP-1074 is a pure antiestrogen in the mouse uterus. While the other analogs of OP-1074 were not tested in the mouse uterine assay, we predict that they would stimulate an increase in wet weight similar to tamoxifen since they exhibited SERM-like activity in the AP assay.

**OP-1074 inhibits proliferation of breast cells and degrades ERα.** In contrast to the uterus, both SERMs and pure antagonists are capable of inhibiting ER-mediated signaling in the breast. Treatment of MCF-7 cells with 100 pM E2 in hormone-depleted media stimulated transcription of a reporter gene containing an estrogen response element (Fig. 2a). OP-1074, OP-1154 and OP-1156 inhibited E2-stimulated transcription in a dose responsive manner with similar IC$_{50}$'s, which were in the low nanomolar range (Fig. 3a). OP-1074 and analogs similarly inhibited E2-induced proliferation in a dose responsive manner to levels below that observed with no E2 stimulation in both MCF-7 and CAMA-1 cells (Fig. 3b, c). Indeed, the potencies for all the analogs, including OP-1039 and OP-1047, were very similar to each other (Table 1), indicating that the relative position of the methyl group on the pyrrolidine does not greatly influence its ability to inhibit ER-mediated signaling in the breast, a tissue in which AF-1 is not required in ER signaling[39,40]. This similarity contrasts with their distinct differences between the analogs in stimulating AP, which behaves as an AF-1 dependent gene.

Recent experimental data indicate that destabilizing ERα expression by increasing receptor turnover is not necessary to achieve pure antagonism in breast cells[5,28,32]. However, it is conceivable that under some clinical contexts SERD activity may still be a desirable feature by reducing the number of functional receptors in tumors. We assessed OP-1074 and analogs for their ability to modulate ERα protein expression in MCF-7 cells by immunoblotting with an ERα specific antibody (Table 1). After a 24-hour treatment with the benchmark SERD fulvestrant, ERα expression declined to 28% of vehicle relative to vehicle in MCF-7 cells. While not as efficacious as fulvestrant, OP-1074 destabilized ERα expression to 49% of vehicle relative to vehicle. A representative immunoblot of treated MCF-7 cells is shown in Fig. 3d along with ERα expression in a second ERα-positive breast cell line, CAMA-1, which showed similar results. In contrast, 2R-, 2S- and 3S- methylpyrrolidines, and the unsubstituted pyrrolidine had no change or even stabilized ERα expression in both cell lines. These data indicate that of the analogs tested in this series, only the 3R- orientation is optimal for receptor degradation.

**OP-1074 increases H12 conformational dynamics.** Since the stereospecific 3 R orientation of the methylpyrrolidine was shown to be necessary for conferring both pure antiestrogenicity and SERD activity in this series, we next explored the molecular interactions with ligand-bound ERα LBD. Relative binding affinities were measured using radioligand-binding assays with [³H]-E2 as a tracer[29]. Overall, the three analogs possessed somewhat

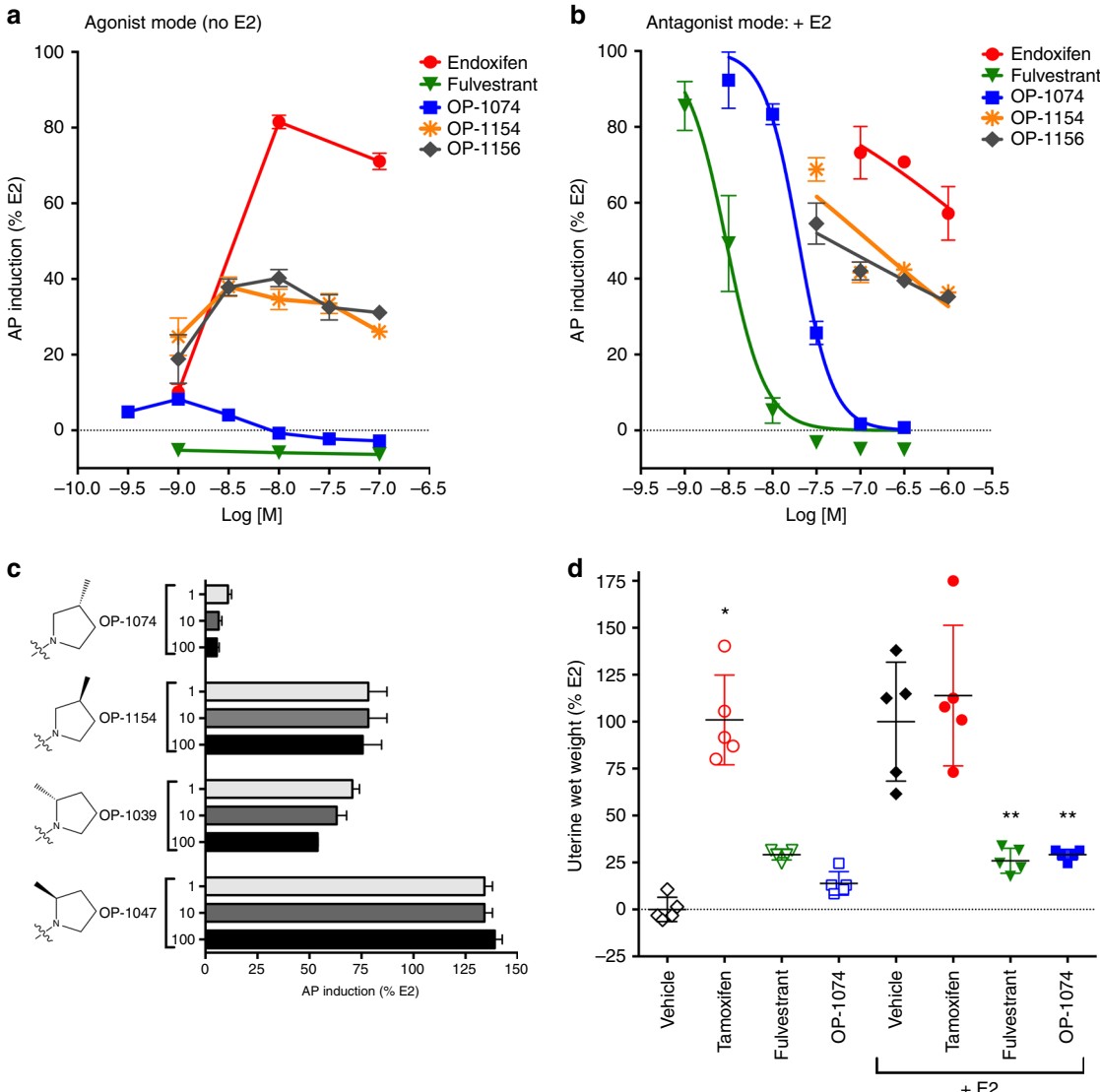

**Fig. 2** OP-1074 is a pure antiestrogen and antagonizes uterotrophic activity. **a** Pure antiestrogens do not stimulate alkaline phosphatase (AP) activity in vitro. Ishikawa cells were treated with compounds for 3 days in the absence of E2 (agonist mode) and assayed for AP activity. Values were normalized for activity of 500 pM E2 in a representative experiment, ±s.e.m., from triplicate wells. **b** Antagonism of E2-stimulated AP activity. Ishikawa cells were similarly treated with compounds as above but in the presence of 500 pM E2 and assayed for AP activity. **c** Only the 3R-position of the methylpyrrolidine group lacks AP activity in the agonist mode. Shown is a representative figure analyzed and presented as above. Note that compounds tested here were mixtures of approximately equal amounts of active and inactive diastereomers. **d** Uterine wet weight measured at the end of 3 days after orally treating ovariectomized BALB/c mice daily with vehicle or 50 mg/kg tamoxifen, 100 mg/ml OP-1074 (mixture of active and inactive diastereomers), or subcutaneously with 50 mg/kg fulvestrant. Half the animals in each group were also administered 0.1 μg/ml E2 subcutaneously. Bars represent mean uterine wet weight ± s.e.m., $n = 5$ animals. *denotes statistical significant from vehicle; **denotes statistical significance from E2 control ($p < 0.05$, one-way ANOVA)

similar binding affinities to each other (Fig. 4a). OP-1074 and OP-1156 bound ER with the highest affinities, with IC$_{50}$'s of 7 and 6 nM, respectively, while affinity for OP-1154 was approximately 2-fold lower. All three bound ER with higher affinity than fulvestrant, which had an IC$_{50}$ of 27 nM. These data indicate that other structural differences within the ERα must be responsible for the PA-SERD properties of OP-1074, rather than merely increased binding affinity.

We found that distinctions between OP-1074, OP-1154 and OP-1156 could be made using an assay that measures the conformational dynamics of ERα LBD H12. This assay employs trypsin challenge fluorescence polarization and was adapted from earlier studies measuring ERα H12 mobility where half-lives ($t_{1/2}$) were calculated using an exponential decay model[29,41]. We found

that labeling position 545 C with tetramethylrhodamine-5-maleimide retained approximately 66% of its affinity compared to WT ERα LBD. The half-life for the unliganded (apo) ERα was 2.2 min, in agreement with previously published data that indicated H12 is highly mobile (Fig. 4b, Supplementary Fig. 3)[29]. When a saturating concentration of E2 was added, $t_{1/2}$ increased to 5.9 min, representing a decrease in H12 mobility. As seen previously, fulvestrant decreased $t_{1/2}$ to 2.3 min (not statistically significant over apo by one-way ANOVA), suggesting that it increased the mobility of H12. The SERM 4-OHT reduced the conformational mobility of H12 with a $t_{1/2}$ of 6.0 min, suggesting that the complex adopted the stable H12 antagonist in the AF-2 cleft, as observed in the x-ray crystal structure[25]. OP-1154 (3S-methyl-pyrrolidine) and OP-1156 (unsubstituted pyrrolidine) did not

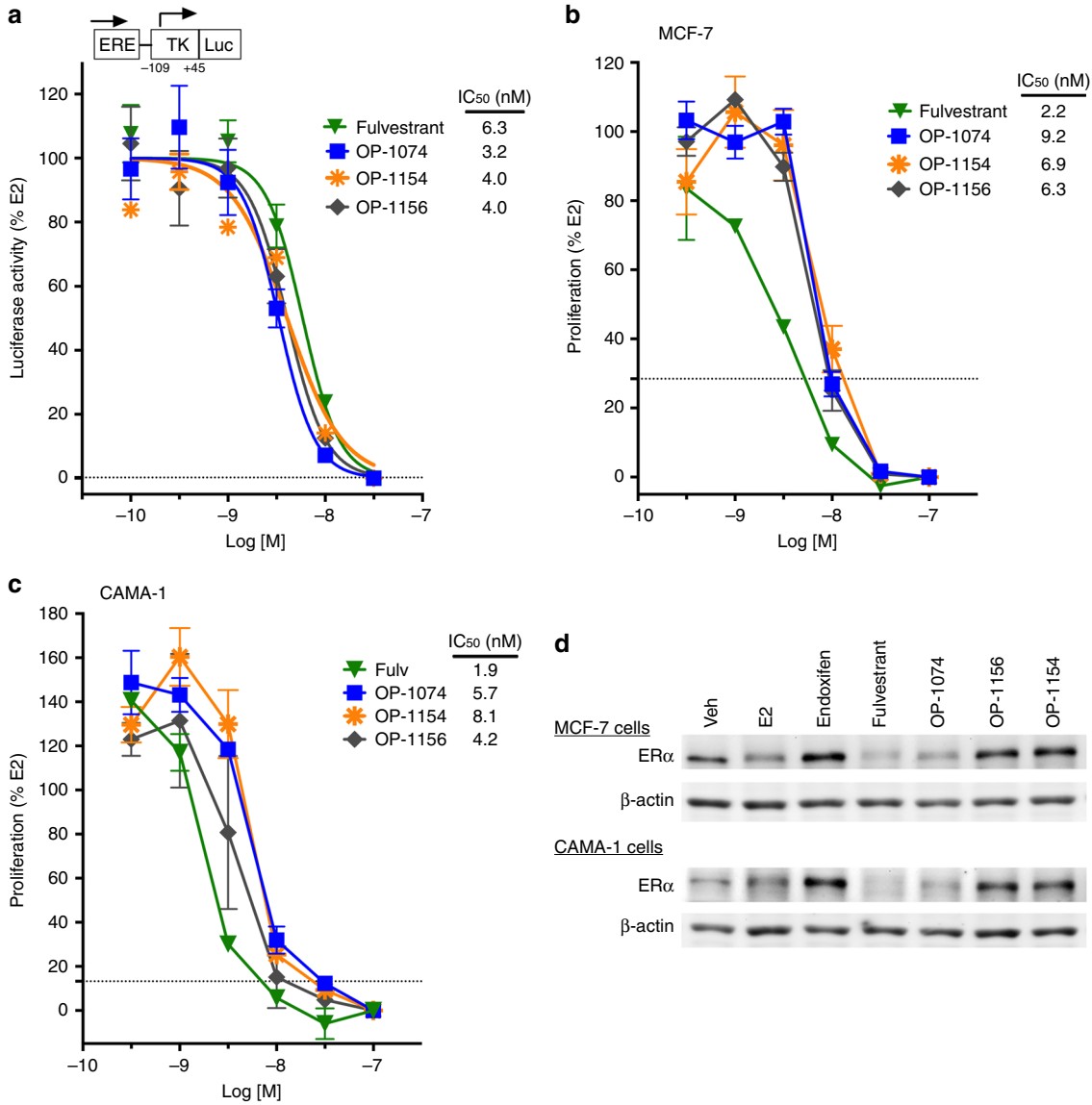

**Fig. 3** OP-1074 inhibits proliferation and degrades ERα in breast cells. **a**, Inhibition of E2-stimulated transcription of an estrogen responsive reporter gene, ERE-tk109-Luc. Luciferase activity was assayed after treating MCF-7 cells with antiestrogens for 22 h in the presence of 100 pM E2. Shown are means +/− s.e.m. from triplicate wells of a representative experiment in which values were normalized to %E2. Dotted line represents basal activity in the absence of E2. **b**, **c**, Inhibition of E2-stimulated proliferation in breast cells. Breast cells were treated with antiestrogens in the presence of 100 pM E2 for 7 days (**b**, MCF-7 cells) or 8 days (**c**, CAMA-1 cells) and subsequently exposed to a fluorescent DNA-binding dye that approximates cell number. Shown is a representative figure analyzed and presented as above. **d**, Western blot of destabilization of ERα protein by OP-1074. MCF-7 and CAMA-1 cells were treated with 100 nM ligands for 24 h and lysates immunoblotted with antibody to ERα and β-actin

show statistically significant differences (by one-way ANOVA) in their half-lives, which were closer to 4-OHT at 4.5 and 4.3 min respectively. In comparison, the $t_{1/2}$ of OP-1074 was 3.5 min; thereby exhibiting increased conformational dynamics of H12 relative to SERM compounds 4-OHT, OP-1154, and OP-1156, but decreased dynamics compared to fulvestrant. Together these data suggest that while OP-1074 may not disrupt H12 to the extent of fulvestrant, it is sufficient to elicit PA-SERD activity in breast cancer cells.

**3R-Methylpyrrolidine of OP-1074 disrupts H12**. High resolution x-ray crystal structures of OP-1154, OP-1156, and OP-1074 were obtained to reveal the structural differences that enable OP-1074 to increase the conformational mobility of H12. The ERα LBD-OP-1154 co-crystal complex was solved to 1.60 Å resolution

by molecular replacement with one dimer in the asymmetric unit (ASU). The ERα LBD-OP-1156 co-crystal complex was solved to 2.00 Å resolution by molecular replacement with one dimer in the asymmetric unit (ASU). The ERα LBD-OP-1074 co-crystal complex was solved to 1.55 Å resolution by molecular replacement with one dimer in ASU. Both OP-1154 and OP-1156 were well resolved in the ligand binding pocket (Fig. 5a–c). Overall, each structure depicts a canonical antagonist binding mode where H12 binds in the AF-2 cleft to block coregulator recruitment. Each molecule participates in similar intermolecular interactions within the hormone-binding pocket by forming hydrogen bonds with its phenolic oxygen and H524, the hydroxyl of the benzyopyran ring and E353, and the nitrogen of the pyrrolidine ring with D351. Van der Waals interactions comprise the remaining interactions with the binding pocket. Fig. 5d–f show the conserved intermolecular interactions between OP-1074 and the

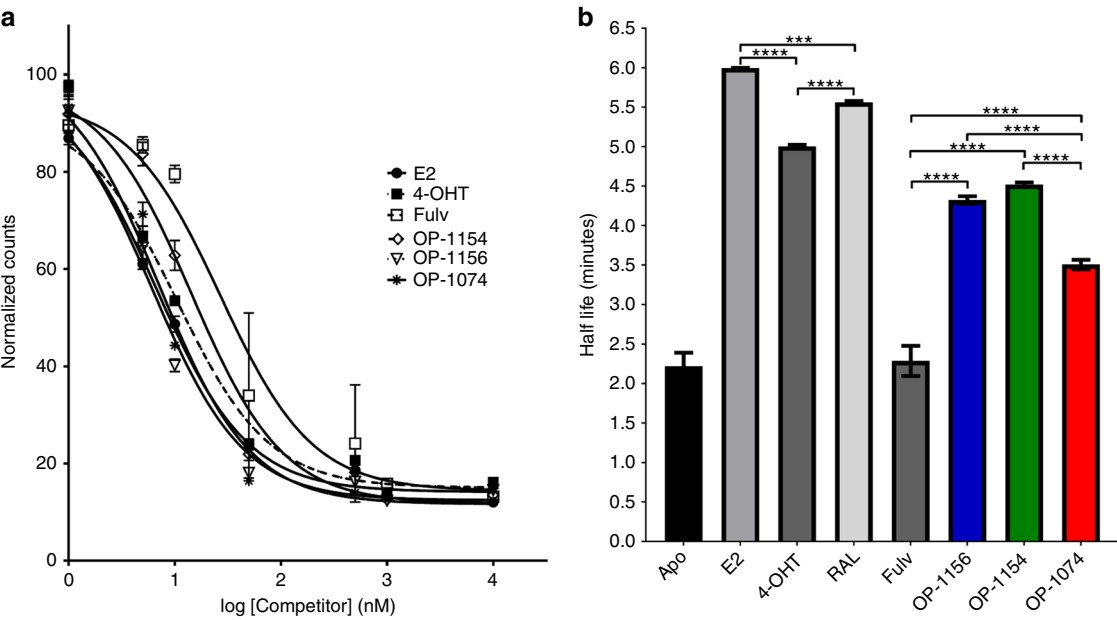

**Fig. 4** Ligand binding affinities and the influence of ligand on H12 mobility. **a**, Competitive binding affinity assay of compounds binding to ERα LBD. Experimental data points are mean $+/-$ s.e.m. of triplicate samples in a representative experiment. Curves were fit using a one-site fit-log model (IC$_{50}$) in GraphPad Prism with $R^2$ of fits were $\geq$ 0.97. The IC$_{50}$ values were 6.71 ± 0.06 nM for E2, 10.46 ± 0.06 nM for 4-OHT, 26.88 ± 0.07 nM for Fulv, 14.3 ± 0.06 nM for OP-1154, 6.13 ± 0.06 nM for OP-1156, and 7.03 ± 0.05 nM for OP-1074. **b**, Measured half-lives of ERα-ligand complexes in a trypsin-coupled fluorescence polarization assay of ERα LBD helix 12 mobility. *Denotes statistical significance between bars (**** for $p < 0.0001$, *** for $p < 0.0001–0.001$) using one-way ANOVA analysis. No significant differences were observed between Apo and fulvestrant or OP-1154 and OP-1156. Experimental data points are mean $+/-$ s.d. of triplicate samples in a representative experiment

hormone binding pocket of ERα LBD that are present in all assayed OP molecules.

The greatest differences between the structures of ERα and the OP compounds lie at the pyrrolidine moiety that lies between H3 and H12. In the ERα-OP-1074 structure, the methyl points towards H12 (Fig. 5f), while it points away from H12 and towards H3 for ERα-OP-1154 (Fig. 5e). OP-1156 does not possess a methyl on its pyrrolidine moiety (Fig. 5d). As such, these structures suggest that the specific direction of this methyl on OP-1074 increases the mobility in the loop connecting helices 11 and 12 (H11-12 loop), which propagates to helix 12. Figure 6 shows a B-factor heat map representations for each structure, which is indicative of conformational mobility within a crystal. B-factors are increased in the H11-12 loop and H12 in the ERα-OP-1074 structure compared to OP-1154 and OP-1156, suggesting that that OP-1074 increases the disorder in the region within the crystal[42]. It should be noted that B-factors were only compared between the "A" monomers/chains of each structure due to significant crystal contacts in the B-chain H12. Together, these structures indicate that the PA-SERD properties of OP-1074 stem from its stereospecific 3R-methylpyrrolidine that disrupts H12 to increase the conformational mobility of the helix.

**Therapeutic potential of OP-1074 in endocrine-resistant models**. The observation that OP-1074 functioned similarly to fulvestrant as a pure antagonist and SERD led us to examine whether it would also exhibit similar efficacy to fulvestrant in an in vivo model of tamoxifen-resistance. In the previously characterized model, MCF-7 breast cells stably overexpressing HER2/neu and implanted into nude mice should eventually become resistant to tamoxifen compared to parental MCF-7 cells, and tumor growth should be inhibited by fulvestrant to similar levels observed with E2 deprivation[43,44]. We observed, in the absence of therapeutic intervention, all tumors continued to grow due to

stimulation by 0.18 mg E2 pellets (90-day release) (Fig. 7a, b). At the end of the study only half of the tumors from animals dosed daily with 100 mg/kg tamoxifen by oral gavage shrunk below the initial tumor size when treatment was initiated. In the absence of prior pharmacokinetic data for OP-1074, animals were treated twice daily with 100 mg/kg drug by oral gavage, whereas fulvestrant-treated animals received daily subcutaneous dosing at the same dose. The extra daily dose may explain some of the enhanced efficacy of OP-1074 compared to fulvestrant observed by day 27 of this study. OP-1074 inhibited the growth of all ten tumors by >50%, and 9/10 tumors shrunk >80%. In comparison, 5/10 of the fulvestrant-treated tumors shrank by >50%, and only 1/10 tumors shrunk >80%. 100 mg/kg of OP-1074 dosed twice a day appeared to be well-tolerated with no adverse effects on behavior or mouse bodyweight. Pharmacokinetic analysis at the conclusion of the study indicated good oral bioavailability of OP-1074 with a C$_{max}$ of 1.3 μg/ml, T$_{max}$ of 1.5 h, T$_{1/2}$ of 3.01 h, and AUC of 8.2 h μg/ml (Fig. 7c).

Recurrent somatic activating LBD mutations have been characterized that are often acquired during prolonged endocrine therapy and are associated with endocrine resistance[45,46]. To see if OP-1074 is efficacious in inhibiting signaling through mutated receptors, we transiently expressed ESR1-Y537S mutant receptor in ER-negative SK-BR-3 cells (Fig. 7d). In a previous study using this model, the ESR1-Y537S mutant promoted hormone-independent transcriptional activation compared to wild-type receptor, and fulvestrant inhibited this activation, although at a reduced potency. We observed that OP-1074 also inhibited this hormone-independent activation induced by the Y537S mutant receptor, with a similar efficacy and potency to fulvestrant. Taken together, data from these 2 models of endocrine-resistant disease suggest that, with adequate dosing, OP-1074 has the potential to be efficacious in shrinking tumors that have acquired resistance to prior endocrine therapy.

**Fig. 5** X-ray crystal structure analysis of OP compounds bound to ERα LBD. **a–c**, Simulated annealing composite omit maps contoured to 1.5 σ for OP-1156 (**a**), OP-1154 (**b**), and OP-1074 (**c**). **d–f**, Hydrogen bonds formed between OP-1156 (**c**), OP-1154 (**d**) and OP-1074 (**e**) and ERα LBD. Helices 3 and 11 are denoted as H3 and H11 respectively. **g–i** Enhanced view of OP-1156, OP-1154, and OP-1074 near the H11-12 loop which ultimately perturbs H12 leading to differential activities between the compounds

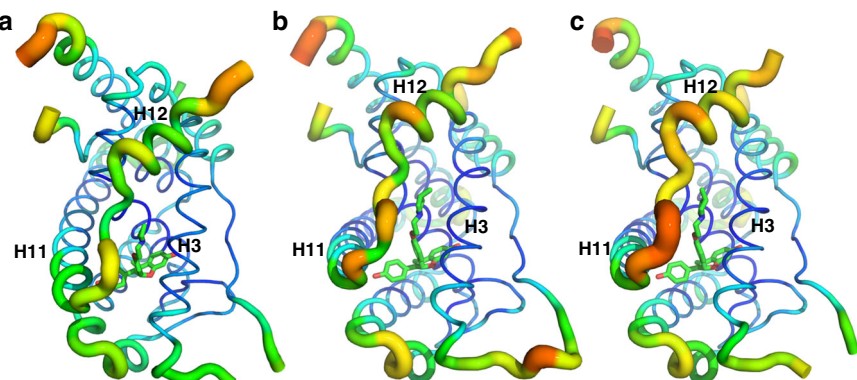

**Fig. 6** OP-1074 increases H12 mobility in the crystal structure. B-factor putty representation of the OP-1156 (**a**), OP-1154 (**b**), and OP-1074 (**c**) compounds in complex with ERα LBD. Cool colors (blues) with small diameter tubes indicate lower B-factors and less mobility in the crystal. Hot colors (reds) with large diameter tubes are indicative of higher B-factors and increased mobility in the crystal. Helices 3, 11, and 12 are labeled H3, H11, and H12 respectively

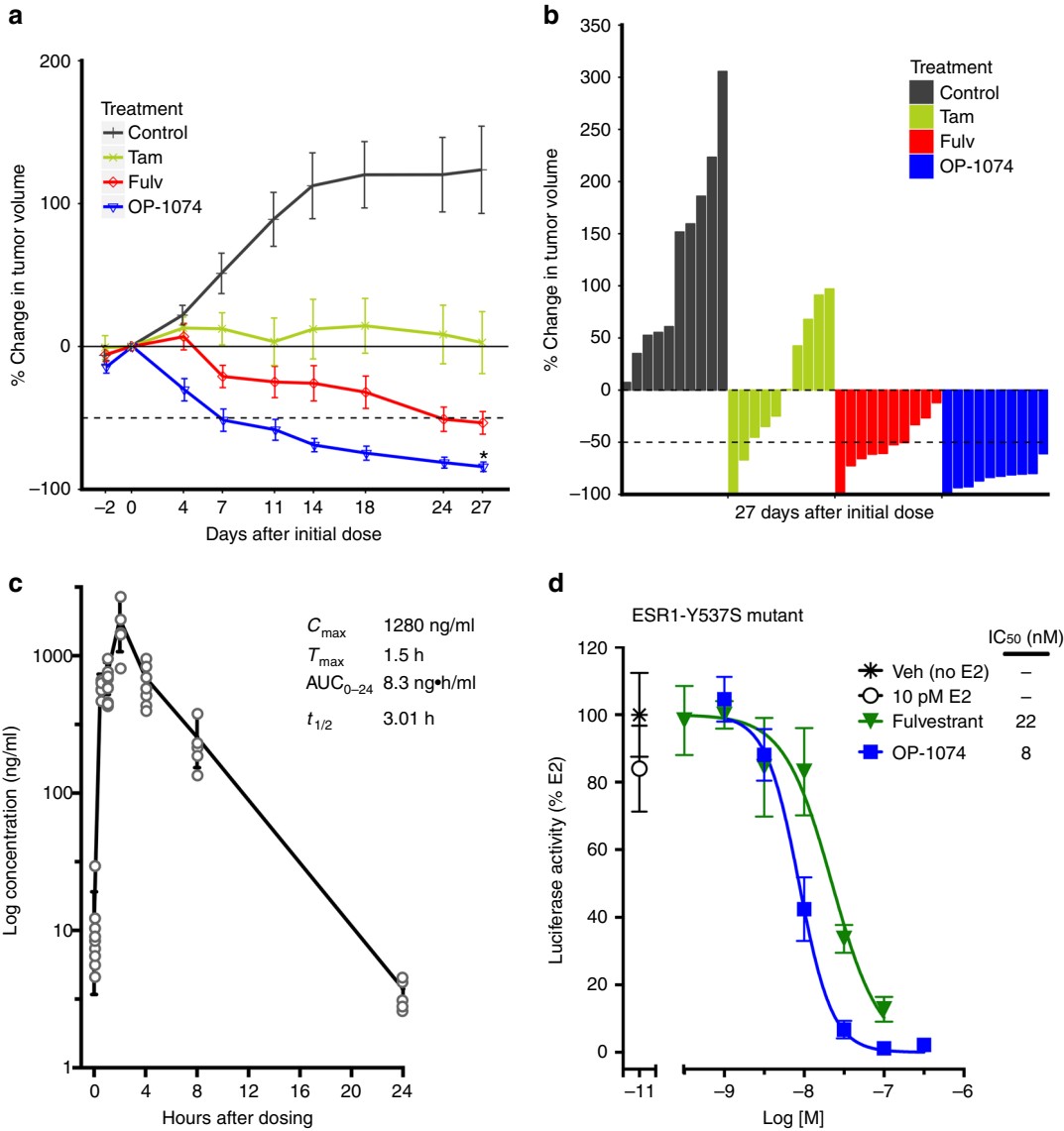

**Fig. 7** OP-1074 shrinks tumors in tamoxifen-resistant xenograft model. MCF7/HER2/neu cells were implanted into ovariectomized athymic nude mice implanted with estrogen pellets. Animals were dosed with 100 mg/kg tamoxifen delivered daily by oral gavage, or 100 mg/kg fulvestrant delivered daily subcutaneously, or 100 mg/kg OP-1074 delivered twice daily by oral gavage. **a** Percent change in tumor volume compared to volume at start of treatment, with each point representing the means±s.e.m., $n = 10$ animals per group. *Denotes OP-1074 is statistically different from fulvestrant at end of study (student's $t$-test, $p < 0.05$). **b** Waterfall plot of percent final tumor volume at end of the study compared with mean volume at start of treatment, with each bar representing one animal. **c** Plot of pharmacokinetic data of OP-1074 measured in mouse plasma on the final dose of OP-1074 over a 24-h period. $n = 10$ animals at time 0, 1, and 4 h. $n = 5$ animals at time 0.5, 2, 8, and 24 h. **d** Inhibition of transcription of LBD mutant ERα by OP-1074. Y537S-ESR1 was transiently transfected into SK-BR-3 cells along with an estrogen responsive reporter gene, ERE-tk109-Luc. Luciferase activity was assayed after treating with ligands for 22 h and values were normalized to %vehicle (no E2) control. Shown are means ± s.e.m. from triplicate wells of a representative experiment

## Discussion

Current efforts to design improved antiestrogens are hindered by a lack of structural information that predicts receptor-ligand conformation(s) required for complete antagonism of ERα. In the present study, a structure activity relationship emerged indicating that the orientation of the 3R- methylpyrrolidine of OP-1074 is sufficient to confer properties of being both a pure antagonist and a SERD (PA-SERD). Biochemical and biophysical assays, coupled with high-resolution x-ray crystal structures, provide a molecular explanation for this activity. OP-1074 increased H12 mobility compared to the SERM-like analogs OP-1156 and OP-1154, which only differ from OP-1074 by an absence and orientation of a methyl group, respectively. While all three analogs displayed

similar binding motifs within the hormone binding pocket, only the 3R-methylpyrrolidine of OP-1074 is directed towards H12. This conformation prevents stable H12 docking in the AF-2 cleft and destabilizes the receptor. In agreement with the trypsin challenge assays, B-factor analysis of the x-ray crystal structures shows that OP-1074 increases the relative disorder of H12 compared to the other analogs.

Although H12 is docked in the AF-2 cleft in the OP-1074 crystal structure, it is likely that the helix is more mobile at physiological temperature, as evidenced by its increased turnover in breast cancer cells and decreased half-life in the trypsin challenge assay. It is interesting that in the structure of another PA-SERD, ICI-164,384, with the rat estrogen receptor beta, H12

cannot be seen in the structure, suggesting that H12 is mobile in the solvent and not docked in the AF-2 cleft[26]. Moreover, when compared to the ERα LBD-endoxifen structure (a SERM), the methylpyrrolidine moiety of OP-1074 adopts a vector that places it closer to H12 than endoxifen's methylamine to disrupt H12 (Supplementary File 4). Thus, the comparison here between the structures of ERα with OP-1074 and structures of closely related, but not pure antagonist compounds, is consistent with the notion that the primary event in regulation of estrogen receptor function by PA-SERDs is destabilization of H12. Exactly how destabilization of H12 leads to degradation of ER and to pure antagonism is an important question that deserves further exploration. Particularly, it should be determined whether H12 instability leads to exposure of amino acids that constitute a degradative signal, and if clearance of the coregulatory cleft allows for corepressor binding as the agent of pure antagonism, as has been suggested[19].

The agonist activity of SERMs in the uterus and resistant breast cells purportedly stems from promiscuous AF-1 activity that can fuel growth factor signaling[7,8]. As such, we used agonism in the uterus as a marker for screening AF-1 activity. Unlike SERMs, OP-1074 lacked agonist activity in uterine cells in vitro and in vivo, and completely antagonized the uterotrophic action of E2. Most importantly, OP-1074 potently inhibited hormone-sensitive and tamoxifen-resistant ER+breast cancer cells in vitro and shrank tamoxifen-resistant tumors in a xenograft model better than fulvestrant. Being a pure antagonist, it will be interesting to see if OP-1074 will share a profile similar to fulvestrant's effects on bone, a tissue in which SERMs have an advantage over pure antagonists by preserving bone mass.

A popular alternate strategy of developing new orally bioavailable ERα antagonists has been to screen ligands for increased turnover of the receptor in order to mimic the SERD activity of fulvestrant. However, recent accumulating clinical evidence suggests that it is fulvestrant's pure antagonism that enables its utility to treat advanced disease, rather than its ability to degrade the receptor[2,47]. Several recent breast cancer candidates in early stage clinical trials, GDC-0810, AZD9694, and RAD1901, all exhibit mild to moderate agonist effects in the uterus and thus are SERM-SERDs[32,48,49]. Because of the promiscuous agonist activity associated with the SERM tamoxifen, we predict that SERM-SERDs will be at a disadvantage in treating and preventing resistant tumors compared to PA-SERDs. Low ER levels mediated by SERD action may yet prove to be a clinical advantage with sufficient dosing under certain circumstances. However, current evidence indicates that complete antagonism, potent binding, and the ability to concentrate drug at the level of the receptor in the breast are the most important features needed to outperform current endocrine therapies.

Mutations in the ER-LBD have been found to be prevalent in relapsed tumors and these somatic mutations cause hormone-independent activation of growth and signaling that confer reduced potencies of antiestrogens[45,46]. We previously speculated that antiestrogens specifically designed to increase the dynamics of H12 will have increased potency against tamoxifen resistant ER+tumors, including those expressing Y537S and D538G[29]. OP-1074 exemplifies our hypothesis by exhibiting considerable potencies, both in tamoxifen-resistant MCF-7 cells and in breast cancer cells expressing the activating ERα Y537S somatic mutation. OP-1074 had good bioavailability in rodents when dosed twice daily at 100 mg/kg dose, and future plans include determining the minimum human equivalent doses necessary to shrink tumors in a variety of resistant settings, which are notoriously heterogeneous and complex. Importantly, our findings inform future design efforts, providing a framework for structure based design for breast cancer drugs with superior antagonist and pharmacological properties.

## Methods

**Ligands.** 17β-estradiol (E2, #E8875), fulvestrant, and (Z)-4-trans-hydroxytamoxifen (4-OHT, #H7904), were obtained from Sigma-Aldrich. (Z)-4-hydroxy-N-desmethyltamoxifen (endoxifen) was purchased from Toronto Research Chemicals (#H938500). Syntheses of OP-1074, OP-1154, OP-1156, OP-1039 and OP-1047 were performed by Synterys, Inc. and methods are fully described in Supplementary Methods. Note that OP-1039 and OP-1047 tested here were mixtures of diastereomers, containing approximately 1:1 active and inactive isomers.

**Cell culture.** MCF-7 cells were obtained from Cheryl L. Walker (Baylor College of Medicine) and maintained in IMEM media (ThermoFisher Scientific, #A10488-01) and supplemented with 10% fetal bovine serum (FBS), (Hyclone, #SH30071). Ishikawa cells were purchased from ATCC (originally designated as ECC-1 cells, now discontinued as they are genetically identical to Ishikawa cells[50]) and maintained in RPMI media (#11835-030) supplemented with 10 mM HEPES (#15630), 1 mM sodium pyruvate (#11360070), all from ThermoFisher Scientific, also supplemented with 10% FBS. CAMA-1 cells were purchased from ATCC and maintained in IMEM media and supplemented with 10% FBS. SK-BR-3 cells were purchased from ATCC and maintained in McCoy's 5 A media (Hyclone, SH30270) and supplemented with 10% FBS. HeLa cells were purchased from ATCC and maintained in DMEM plus 10% FBS. Cell lines were authenticated by short tandem repeat DNA profiling by Laragen in January, 2017. Cell lines were determined by PCR to be free of mycoplasma by Laragen in January, 2017.

**Alkaline phosphatase (AP) assay.** Approximately 15,000 Ishikawa cells per well were plated into a 96-well plate in media containing 5% charcoal dextran stripped FBS (HyClone, #SH30068). At least 4 h later cells were treated with antiestrogens and media was diluted to 2.5% stripped FBS. Cells were incubated for 3 days, media removed and plates frozen at −80 °C. Thawed plates were incubated with a chromogenic substrate of AP, p-nitrophenyl phosphate (ThermoFisher Scientific, #36721), for 40–60 min at 40 °C, and absorbance read at 405 nm on the Varioskan Lux multimodal plate reader (ThermoFisher Scientific, #VLBL00D0). AP activity in the antagonist mode was assayed as above but cells were co-treated with 500 pM E2. Data shown in figures are representative experiments reproduced at least three times in independent AP assays.

**Cell proliferation assays.** In total 900 cells per well were plated into a 96-well plate in media plus 5% charcoal dextran stripped FBS. At least 4 h later cells were treated with antiestrogens and media was diluted to 2.5% stripped FBS in the presence of 100 pM E2 for 6-8 days. Proliferation was measured using CyQuant fluorescent DNA-binding dye kit (ThermoFisher Scientific, #C7026) using 1:200 GR dye and reading fluorescence at 485 nm excitation and 538 nm emission on the Varioskan Lux multimodal plate reader. Data shown in figures are representative experiments reproduced at least three times in independent proliferation assays.

**Luciferase assay.** Approximately 45,000 MCF-7 cells per well were transiently transfected with 70 ng of an estrogen-responsive reporter gene, ERE-tk109-Luc[18], using Lipofectamine LTX (ThermoFisher Scientific, #15338). Transfected cells were plated into a 96-well plate in media containing 5% charcoal dextran stripped FBS. At least 4 hours later cells were treated with antiestrogens diluted to 1.25% stripped FBS in the presence of 100 pM E2 for 22 h. 50 μl Bright Glo luciferase reagent (Promega, #E2620) was used to lyse cells and detect firefly luciferase activity, measured in relative light units, at 1 s intervals using the Varioskan Lux multimodal plate reader. SK-BR-3 cells were transfected as above but with 10 ng Y537S-ESR1[19]. Method for assessing specific transcriptional activity of OP-1074 on different steroid hormone receptors is described in the Supplementary Methods. Data shown in figures are representative experiments reproduced at least 3 times in independent luciferase assays.

**Immunoblotting.** Approximately 350,000 cells per well were treated with 100 nM antiestrogen for 24 h in serum-free medium in 12-well plates. Cells were lysed with RIPA buffer (#89901) supplemented with a cocktail of protease and phosphatase inhibitors (#78410), both from ThermoFisher Scientific. Whole protein extracts were separated on 10% SDS–PAGE TGX gels (#5671035) and transferred to nitrocellulose membranes (#1704159), both from BioRad. Blots were incubated with either 1:1000 dilution of D12 ERα (#sc-8005), or 1:2000 dilution of β-actin (#sc-47778) primary antibodies, followed by 1:2000 dilution of goat anti-mouse-HRP secondary antibody (#sc-2005) (all antibodies from Santa Cruz Biotechnology). Signal was detected with Super Signal Femto chemiluminescent reagent (ThermoFisher Scientific, #34095) and ERα bands normalized to β-actin to control for differences in loading. Imaging was performed on FluorChem Imager by Alpha Innotech. Data shown in figures are representative experiments reproduced at least three times in independent Western assays. An uncropped image of the immunoblot is shown in Supplementary Fig. 5.

**Uterine wet weight.** Uterine wet weight experiment was conducted once at the Preclinical Therapeutics Core at the University of California, San Francisco

following the ethical regulations of institutional animal care and use committee (IACUC) protocols (study #PTC-1202). Ovariectomized female adult BALB/c mice (Taconic Biosciences) were randomly assigned to 8 treatment groups, 5 animals per group, that were treated once daily with vehicle (0.5% carboxymethylcellulose (CMC)), or one of the following treatments: 50 mg/kg tamoxifen in CMC, 50 mg/kg fulvestrant in 5% ethanol (subcutaneous), or 100 mg/ml OP-1074 in 0.5% CMC (oral gavage). Half the animals in each group were co-treated with 0.1 µg/ml E2 delivered subcutaneously in cottonseed oil-ethanol (95:5), or vehicle alone. Animals were euthanized and uteri dissected and weighed.

**Murine xenograft study**. Xenograft study was conducted once at the Preclinical Therapeutics Core at the University of California, San Francisco following the ethical regulations of IACUC protocols, (study #PTC-1217). MCF7/HER2/neu cells were grown in culture and implanted into 6–7 week athymic ovariectomized female nu/nu mice (Taconic Biosciences). To stimulate tumor growth, 0.18 mg estradiol 90 day release pellets (Innovative Research) were implanted along with cells. When tumors reached ~250 mm³ animals were randomly divided into four groups (10 mice per group): 100 mg/kg tamoxifen in 0.5% CMC (oral gavage) and treated daily, 100 mg/kg fulvestrant in 5% ethanol (subcutaneous) and treated daily, or 100 mg/ml OP-1074 in 0.5% CMC (oral gavage) and treated twice daily, except on weekends/holidays when they were treated once daily. Tumor volume and body-weight were measured twice weekly. At end of the study animals were euthanized, tumors resected and snap frozen in liquid nitrogen, and uteri were resected, weighed, and fixed in 10% formalin.

**Competitive ligand binding assay**. In total 2 nM biotinylated Estrogen Receptor α LBD (ERα) was incubated with serial dilutions of ligands in TBS-T with 10 nM tritiated estradiol ([³H]-E2, Perkin Elmer #NET317250UC) and 10 mM dithio-threitol (DTT). Final concentrations of [³H]-E2 competitors ranged from 0.1 nM to 10 µM. Triplicate reactions were vortexed and 100 µL was added in triplicate to a 96-well streptavidin flashplate (Perkin Elmer #SMP103001PK), which incubated overnight at 4 °C prior to reading. Active ERα bound [³H]-E2 was detected with a Wallac 1450 Microbeta scintillation counter. Data were normalized to their maximum count value prior to plotting and all $R^2 \geq 0.97$.

**Trypsin proteolysis**. A pET21a(+) vector containing a hexa-His-TEV fusion of the ERα LBD (residues 300-550) with each solvent exposed cysteine changed to serine (C381S, C417S, C530S) was synthesized by GenScript. Based on ERα LBD x-ray crystal structures, Q5 site-directed mutagenesis (New England Biolabs #E0554S) was used to incorporate cysteines at solvent exposed position D545C using the following primer:
*D545C Forward*: (5′GGAAATGCTGTGCGCGCACCGTC3′)
*D545C Reverse*: (5′AGCAGCAGGTCATACAGC3′). The sequence for the resulting ERα LBD mutants was verified by sequencing. A 250 mL LB broth containing 100 µg/ml ampicillin was inoculated with a single colony of the *E. coli* expression strain BL21(DE3) transformed with the pET21a(+) ERα LBD mutant. Following overnight incubation at 37 °C with shaking, 3 × 1 L LB-amp broths were inoculated with 5 ml of the overnight culture. Cells grew at 37 °C with shaking at 225 rpm until they reached mid-log-phase growth ($OD_{600} = 0.8$), at which point protein expression was induced using 0.3 mM IPTG and expression continued overnight at 18 °C. Cells were harvested by centrifugation at 3500 $g$ for 30 minutes. Fresh pellets were re-suspended in 200 mL of buffer containing 20 mM Tris pH 8.0, 500 mM NaCl, 1 mM DTT, 5% glycerol, 40 mM imidazole pH 8.0. Protease inhibitor cocktail (Roche #04693159001) and lysozyme (Sigma #62970) were added to the pellets and cells were fully resuspended by mixing at 4 °C for 1 hr. Cells were further lysed by sonication and the mixture was centrifuged at 22,000 $g$ for 30 minutes at 4 °C and the supernatant was isolated. The soluble fraction was incubated with 2 mL pre-washed Ni-NTA resin (ThermoFisher Scientific #88221) then placed onto a column. The column was washed with 10 column volumes of buffer containing 20 mM Tris pH 8.0, 500 mM NaCl, 1 mM DTT, 5% glycerol, 40 mM imidazole pH 8.0. The protein was eluted from the column using a buffer containing 20 mM Tris pH 8.0, 500 mM NaCl, 1 mM DTT, 5% glycerol, 500 mM imidazole pH 8.0. The protein was concentrated using spin concentrators with 30,000 molecular weight cutoff (EMD Millipore #UFC903008) then purified over a Superdex 200 HiLoad 200 16/600 size exclusion column (GE Life Sciences #28989335) with a buffer containing 20 mM HEPES pH 7.4, 150 mM NaCl, 50 µM DTT, and 5% glycerol.

Protein labeling was performed by incubating the single solvent exposed cys mutant with 10-fold excess tetramethylrhodamine-5-maleimide (Sigma Aldrich #94506) for 24 h at 4 °C. This label is an effective indicator of conformational flexibility using fluorescence polarization[29]. Excess fluorophore was removed using dialysis for 72 h with 10,000 molecular weight cutoff Snakeskin dialysis membrane (ThermoFisher Scientific #68100). Activity of the 545 C ERα LBD mutants was measured using our [³H]E2 binding assay. For the trypsin-challenge fluorescence polarization assay 20 nM labeled protein was incubated with 20 nM non-labeled protein overnight to reduce homo-FRET artifacts[29,41]. The protein was diluted to 10 nM in buffer containing 20 mM HEPES pH 7.4, 150 mM NaCl, 50 µM DTT, and 5% glycerol and was incubated with 1 µM ligand for 16 hours at 4 °C. 270 µl of protein-ligand mixture was added in triplicate to a Costar 96-well black

opaque plate (Corning #3792) and trypsin was added to a final volume of 300 µl to yield a final concentration of 100 ng/ml. Fluorescence polarization was measured at 0, 10, 30, 60, 120, and 300 min with a Synergy NEO 2 HTS Plate Reader (BioTek) using the fluorescence polarization (FP) filter cubes 4 (dual FP) and 62 (530/590). Gain was automatically adjusted using a well that contained 10 nM tetramethylrhodamine-5-maleimide in buffer (no protein). All FP signals were at least 10-fold greater than the fluorophore-alone control. Curves were fit using an exponential decay model in GraphPad Prism. Data were normalized to the non-trypsinized control wells at each time point. The $R^2$ for the quality of the fit was at least 0.90 for all fits. One-way ANOVA in GraphPad Prism was used to determine significance between half-lives.

**Protein expression and purification for x-ray crystallography**. A gene containing a hexa-His-TEV fusion of the ERα LBD (residues 300-550) was synthesized by GenScript containing C381S, C417S, C530S, and L536S mutations in pET21(a) + [48]. For protein expression, a 250 mL LB broth containing 100 µg/mL ampicillin was inoculated with a single colony of the *E.coli* expression strain BL21(DE3) transformed with the pET21a(+) ERα LBD mutant. Following overnight incubation at 37 °C with shaking, 10 × 1 L LB-amp broths were inoculated with 5 mL of the overnight culture. Cells grew at 37 °C with shaking at 225 rpm until they reached mid-log-phase growth ($OD_{600} = 0.8$), at which point protein expression was induced using 0.3 mM IPTG and expression continued overnight at 18°. Cells were harvested by centrifugation at 3500 $g$ for 30 min. Fresh pellets were re-suspended in 200 mL of buffer containing 20 mM Tris pH 8.0, 500 mM NaCl, 1 mM DTT, 5% glycerol, 40 mM imidazole pH 8.0. Protease inhibitor cocktail and lysozyme were added to the pellets and cells were fully re-suspended by mixing at 4 °C for 1 h. Cells were further lysed by sonication and the mixture was centrifuged at 22,000 $g$ for 30 min at 4 °C and the supernatant was isolated. The soluble fraction was incubated with 2 mL of pre-washed Ni-NTA resin then placed onto a column. The column was washed with 10 column volumes of buffer containing 20 mM Tris pH 8.0, 500 mM NaCl, 1 mM DTT, 5% glycerol, 40 mM imidazole pH 8.0. Protein was eluted from the column using a buffer containing 20 mM Tris pH 8.0, 500 mM NaCl, 1 mM DTT, 5% glycerol, and 500 mM imidazole pH 8.0. The protein was separated from the hexa-His tag by incubation with a 15:1 w/w ration of LBD to hexa-His-TEV protease. The LBD was isolated and the tag removed by passing the protein over pre-washed Ni-NTA beads. Protein was concentrated to 5 mL using a spin concentrator at 4 °C then further purified using Superdex 200 HiLoad 200 16/600 size exclusion column equilibrated with 20 mM Tris pH 8.00, 150 mM NaCl, 1 mM DTT, and 5% glycerol. A single peak was observed on the chromatogram. Fractions corresponding to that peak were pooled, concentrated to 20 mg/ml then flash frozen and stored at −80 °C for later use.

**Crystallization of ERα LBD in complex with benzopyran ligands**. The purified LBD was incubated with 2 mM of each compound for 4 h on ice prior to crystal screens then centrifuged at 20,000 $g$ for 30 min at 4 °C to remove insoluble ligand. Hanging drop vapor diffusion where 2 µL protein was mixed with 2 µL well solution using VDX plates (Hampton #HR3-140) was used to generate all crystals. For OP-1156, clear star-shaped crystals emerged from the drops after 5 days at 10 mg/mL protein concentration in 15% PEG 3350, HEPES pH 6.5, and 200 mM MgCl₂. For OP-1154, clear star-shaped crystals formed after 4 days in 20% PEG 3,350, Tris pH 7.5, and 200 mM MgCl₂. For OP-1074, clear rhombohedral crystals formed after 7 days in 10% PEG 3,350, Bis-Tris pH 6.2, 200 mM MgCl₂. 25% glycerol mixed with mother liquor was used as a cryo-protectant for all crystals. All x-ray data sets were collected at the Advanced Photon Source Argonne National Laboratories, Argonne, Illinois on the SBC 19-BM beamline (0.97 Å). Supplementary Table 1 contains a table with crystallization data collection and refinement statistics.

**X-ray structure solution**. Data were indexed, scaled, and merged using HKL-3000[51]. Phenix was used for all molecular replacements using PDB: 5ACC as a search model after removing all ligands and waters[48,52]. Phenix Refine was used for refinement[52]. Models were refined using iterative rounds of Phenix refine and manual inspection with Coot[52]. Densities for OP-1074, OP-1154, or OP-1156 were clearly visible after one round of refinement. Elbow was used to generate the constraints for each compound[52]. Unresolved residues were not included in the final model. All x-ray crystal structure images were made using Pymol. Supplementary Figure 6 shows a stereo-view of the electron density map for OP-1156 bound to ERα LBD. Supplementary Figure 7 shows a stereo-view of the electron density map for OP-1156 bound to ERα LBD. Supplementary Figure 8 shows a stereo-view of the electron density map for OP-1074 bound to ERα LBD.

**Statistical analysis**. Data were analyzed and graphed using Graphpad Prism v5.1 or v6.0 except for Fig. 7a,b written in R Software using ggplot2 library. Dose response curves were fit using the least squares fit method and $IC_{50}$'s calculated using a variable slope sigmoid dose-response model. Where reported, statistical significance was determined by Student's t-test (two-tailed) (Fig. 7a), or using one-way ANOVA followed by Dunnett's multiple comparisons test (Figs. 2d, 4b). $p$-values ≤ 0.05 were considered statistically significant and are stated for each figure. Sample sizes were chosen without consideration of the power needed to detect a

pre-specified effect size. Cell-based in vitro experiments were repeated at least three times to confirm results. Crystal structures and animal based studies were performed once. Animals were randomly assigned to treatment groups using program StudyLog. Investigators were not blinded with regard to treatment in any of the experiments. For the trypsin-coupled fluorescence polarization assays, curves were fit using an exponential decay model in GraphPad Prism (Supplementary Fig. 3). Data were normalized to the non-trypsinized control wells at each time point. The $R^2$ for the quality of the fit was at least 0.90 for all fits. One-way ANOVA in GraphPad Prism was used to determine significance between half-lives (Fig. 4b). $p < 0.001$ were considered statistically significant in this assay. All statistics relevant to the x-ray crystal structures are found in Supplementary Table 1.

**Data availability**. X-ray structures were deposited in the protein databank (PDB) with accession codes 5UFW for ERα LBD-OP-1154, 6C42 for ERα LBD-OP-1156, and 5UFX for ERα LBD-OP-1074. The data that supports this work is available from authors upon reasonable request.

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

## Acknowledgements

X-ray crystal structure data sets were collected at Argonne National Laboratory, Structural Biology Center at the Advanced Photon Source. Argonne is operated by UChicago Argonne, LLC, for the US Department of Energy, Office of Biological and Environmental Research under contract DE-AC-02-06CH11357. We thank Byron Hann at University of California San Francisco for guidance on vivo experiments. We thank the team of chemists at Synterys, Inc. for their guidance on chemical synthesis. Financial support: Susan G. Komen Foundation PDF14301382 (to S.W.F.); Virginia and D.K. Ludwig Fund for Cancer Research (to G.L.G.).

## Author contributions

L.H.-G., S.W.F., P.J.K., G.L.G., C.L.H., and D.C.M. conceived, designed, and coordinated research. L.H.-G. S.W.F., C.E.F., B.D.G., I.N.P., and R.S. were involved in performing in vitro experiments, including developing methodologies and analyzing results. D.C.M. designed, coordinated and guided the chemical syntheses. All authors were involved in interpreting data results. L.H.-G. and S.W.F. wrote the manuscript with edits contributed by D.C.M., P.J.K., G.L.G., C.L.H., R.S., and D.C.M.

## Additional information

**Competing interests:** L.H., D.C.M., R.S., C.L.H., G.L.G., and P.J.K. have financial interests in Olema Pharmaceuticals, which holds a patent for OP-1074 (USPTO 9018244).

