## [Peer Review File · Nature Communications]

Reviewers' comments:

Reviewer #1 (Remarks to the Author):

In the manuscript, the authors synthesized five benzopyrans with variable side chains to identify pure antiestrogens in uterotrophic assay. Compound OP-1074 was identified as a pure antiestrogen and selective ER degrader. Oral administration of OP-1074 proved to be more efficacious than fulvestrant in shrinking tumors in a tamoxifen resistant xenograft model. Further biochemical and crystal structure analyses revealed a structure activity relationship implicating the importance of a stereospecific methyl on the pyrrolidine side chain of OP-1074. The work is indeed of interest, and systematical evaluations were conducted, revealing that compound OP-1074 does exhibit better properties than drug fulvestrant. However, I do not see it as being novel, readership, or of the impact requisite for publication in Nature Communications. My recommendation would be that it would be more suitable to a more specific-focused journal.

Other comment : The library of benzopyran compounds is small. The authors stated that methylpyrrolidines in the side chain was crucial to the reactivities. What about other substituents rather than methyl group? What about other nitrogen-containing moieties instead of pyrrolidine? A screen for a larger library would make the study more interesting and make the conclusion more solid.

Reviewer #2 (Remarks to the Author):

This manuscript describes a new antagonist (OP-1074) for estrogen receptor alpha ($ER\alpha$). The authors claim that the specific positioning of a methyl residue on the pyrrolidine side chain of OP-1074 affects to the movement of the helix 12 (H12) in the ligand binding domain of $ER\alpha$ coincides with the $ER\alpha$ protein degradation.

There is no figure number and these figures are arranged in random order in pdf that makes difficulty to read this manuscript.

In this report the authors compared activities of OP-1074, OP-1154 (enantiomer) and OP-1156 (unsubstituted) in several experiments to demonstrate correlation between the position of methyl substituted pyrrolidine ring and biological activity. These are reasonable experiments. The

authors concluded that the 3R methyl substituted pyrrolidine ring of OP-1074 induces ER α antagonist activity using the alkaline phosphatase (AP) assay in Ishikawa cells and mouse uterine growth assay (Fig.1). Furthermore, the authors showed that OP-1074 reduced ER α protein level in the MCF7 cells (Fig.2E). From these results, the authors concluded that the selected estrogen receptor degradator (SERD) activity is the antagonist activity of OP-1074. However, there is no differential activity of those three chemicals for MCF7 cell proliferation and reporter assay (Fig.2A&B). It makes a question for the antagonist mechanism of OP-1074 which the author's claim. The effect of 3R methyl substituted pyrrolidine ring of OP-1074 might affect only the AP activity in Ishikawa cell but not other biological responses. Regrettably, there are no information of OP-1154 and OP-1156 effects on uterine growth and other breast cancer cell growth. Moreover, there is no information of OP-1154 and OP-1156 effect on ER α protein level in the cells except MCF7 cells. There is less information to evaluate their conclusion. The authors have to add the results of OP-1154 and OP-1156 for uterine growth assay (Fig.1D), MCF7-Her2/neu proliferation assay (Fig.2C) and ESR1-Y537S mutant reporter assay (Fig.2D). Furthermore, the ER α level in the OP-1154 and OP-1156 treated cells (Ishikawa, CAMA-1 and ZR-75-1 cells). The authors should reconsider their conclusion with those results.

The authors performed tamoxifen resistant MCF7 xenograft experiment to evaluate the efficacy of anti-cancer effect of OP-1074 in vivo compared to tamoxifen and fulvestrant. They concluded that the OP-1074 repressed the growth of tamoxifen resistant MCF7 cells more effectively than tamoxifen or fulvestrant (Fig.3). However, they treated OP-1074 twice daily in spite of others were once daily. This experimental design is quite questionable to describe their conclusion. To prevent misunderstanding of readers, the authors should emphasize this different treatment scheme. Or the authors should reevaluate the results using the same treatment scheme. Moreover, the authors need to show experimental data of pharmacokinetic analysis (p.10).

To verify the chemical dependent H12 movement, the authors employed Trypsin-coupled fluorescent polarization assay. The experimental results are shown in supplemental file S4. This result is important for suggesting the OP-1074 enhanced H12 movement. However, the result is unclear to convince this reviewer. If OP-1074 has similar activity to fulvestrant, profile should be similar to fulvestrant (ICI) or Apo in fig.S4. However, the profile of OP-1074 is similar to other OP chemicals instead of fulvestrant or Apo. It is hard to understand why the X axis of fig S4 using the log₂ scale. It is just 3, 4, 5, 6, 7, 8 min. How did the authors evaluate Apo and fulvestrant mediated half-life (2.2 min and 2.3 min respectively) from this experiment? They have to show us understandable results.

minor

The authors analyzed agonist and antagonist activity of OP-1074 and described in the results “OP-1074 lacked activity on other nuclear receptors and...”. This sentence should be “OP-1074 lacked activity on other steroid hormone receptors and...” (p.6).

The authors show IC50 in the figures as log10 (there is no description). It is unkind for the readers. It should be described as nM (Fig. 2 and Fig.s2).

There is no endoxifene (Endox) result in Fig2B and FigS3 despite it is described in the text. The authors have to show the Endox results to compare the results of Fig.1.

There is no information of treated chemical concentration for ER α western blot analyses (Fig2E, FileS3C, D). The authors should describe information. There is unrelated experiment statement in supplement figure S3 legend.

The label of X axis in Fig.4A is not proper. It should be log (nM).

Reviewer #3 (Remarks to the Author):

In this manuscript, Fanning et al. analyze the activity and structure of a new pure ER antagonist, OP-1074, and 2 analogs. Pure ER antagonists have potential as breast cancer therapeutics for patients who have developed resistance against the tissue specific (SERM) antagonist tamoxifen, as shown for the clinically approved pure antagonist Fulvestrant. The authors previously patented OP-1074, which has a 3R-methylated pyrrolidine moiety, and designed 4 analogs, of which they've tested two: OP-1154, which is the 3S substituted analog, and OP-1156, which lacks a methyl group in the pyrrolidine moiety. They demonstrate that OP1074, but neither OP-1154 nor OP-1156, behaves as a full antagonist in an alkaline phosphatase activity assay and a breast cancer cell line proliferation assay, although OP1074 is less potent than Fulvestrant. In a xenograft model, OP-1074 delivered 2x daily caused stronger tumor shrinkage than Fulvestrant delivered 1 x daily, which is therapeutically important since OP-1074 has better bioavailability than Fulvestrant, which needs to be delivered by intramuscular injection.

A structural hallmark of ER agonists is that they reduce H12 dynamics, whereas the authors show that OP-1074 increases H12 dynamics, as expected for a reverse agonist. To understand its structural basis, the authors determined the crystal structures of the ER α ligand binding domain in complex with OP-1074, OP-1154, and OP-1156. While all three structures show the hallmarks of an antagonist conformation, the 3R methyl group of OP-1074 points more toward H12, while the 3S methyl group of OP-1154 points more towards H3. Together with an increased B-factor in the H11-H12 linker and also in the part of H12 close to the methyl group, the authors take this “correlation” as indication that OP-1074 functions as pure antagonist/reverse agonist by

destabilizing H12 due to the ability of the 3R-methyl to disrupt H12.

Overall, the description and analysis of the antagonist OP-1074 and the model of H12 disruption by the stereospecific methylpyrrolidine group are novel and of interest to the field. Although the authors stop short of providing definitive evidence, the data are clearly in support of this interesting model. However, there are number of issues that need to be addressed before I can recommend the paper for publication:

1. There are several instances of overreaching or incorrect statements. In the paper that the authors cite for the ability of the AP assay to discriminate between pure antiestrogens and SERMs, this assay classified the bioactive tamoxifen metabolite monohydroxytamoxifen as pure antagonist, even so tamoxifen functions as a SERM. Similarly, endoxifen is not “the” bioactive metabolite of tamoxifen, but rather “one” of the metabolites, and the authors switch between endoxifen in the AP assay and 4-hydroxytamoxifen (4-OHT) in other assays. Ideally, the authors should include both metabolites in all in vitro assays. Similarly, the statement that “OP-1074 was slightly less potent than Fulvestrant, and more potent than 4-OHT” is biased as the potency was exactly in the middle between that of Fulvestrant and 4-OHT.
2. The authors should clarify that pure antiestrogens are not always advantageous over SERMs. Rather, tamoxifen has remained the first line treatment of ER-positive breast cancer, its agonist activity in bone is clearly beneficial, and endoxifen has been found to be effective in patients who have failed Fulvestrant treatment.
3. Fig. S3B: Please show actual data, rather than just a summary table.
4. Please label the main helices and loops in Figs. 5 and 6.
5. Given the proposed critical role of the stereospecific methylpyrrolidine group, the authors should prepare a figure to show the interaction of the methyl group with the receptor residues. Also, a figure for the comparison of OP-1074 complex with that of tamoxifen/hydroxyl tamoxifen will be helpful.
6. In panel D of Figure 5, the distance of 1.8 Å for a hydrogen bond is too short. The R free and R factors for the OP-1156 complex are a little too high based on its resolution of the data. Further refinement is needed to optimize the geometry of the model.
7. Suppl File S5: the authors should include the high resolution shell data of Rmerge, completeness, and redundancy. Again, the R factors of OP-1156 complex are a little high based on the 2 Å resolution of the data.

Responses to Reviewers:

It is our pleasure to submit our revised manuscript for consideration to *Nature Communications*. We would like to sincerely thank the Editors and Reviewers for considering our manuscript and providing constructive comments. We have addressed these comments and have made the following changes to the previously submitted manuscript:

- Reviewer 1 asked about other substituents besides the methyl on the pyrrolidine moiety.
 - o The purpose of this manuscript was not to do a large screen of compounds, but rather, to focus on understanding why the orientation of the methyl group changes the action of the compound. Towards that end, we have included a clarifying sentence in the introduction: “Our purpose in designing these analogs was to explore the stereochemical space of the antiestrogen side-chain shown to be involved in disrupting the active LBD conformation and occlude coactivator binding by repositioning H12.” We believe that that the dramatic differences found between the analogs represent an important advance about the optimal orientation of ligand binding to ER α to induce complete antiestrogenicity, and informs future drug design for treating endocrine-resistant tumors.
- Reviewer 2 stated: “There are no figure numbers and the figures are arranged in random order.”
 - o This must have been an issue with formatting. The compiled manuscript appeared correct to us. We will triple-check to make sure that the revised manuscript is correctly compiled upon resubmission.
- Reviewer 2 stated: “However, there is no differential activity for those three chemicals (OP compounds) for MCF7 cell proliferation and reporter assay. It makes a question for the antagonist mechanism of OP1074 of which the authors claim. The effect of the 3R methyl substituted pyrrolidine ring might affect only the AP activity in the Ishikawa cell but not other biological responses.” Moreover, there is no information of OP-1154 and OP-1156 effect on ER α protein level in the cells except MCF7 cells. There is less information to evaluate their conclusion. The authors have to add the results of OP-1154 and OP-1156 for uterine growth assay (Fig.1D), MCF7-Her2/neu proliferation assay (Fig.2C) and ESR1-Y537S mutant reporter assay (Fig.2D). Furthermore, the ER α level in the OP-1154 and OP-1156 treated cells (Ishikawa, CAMA-1 and ZR-75-1 cells). The authors should reconsider their conclusion with those results.”
 - o Within the text we highlighted that the in vitro AP assay robustly predicts in vivo results, which allows ligands to be quickly screened using considerably fewer resources. Another sentence was added that based on the other analog’s SERM-like activity in the AP assay we predicted that they would induce an increase in rodent uterine wet weight. We agree with the concern that not all the analogs were tested in a second breast cell line so we have added two new experiments, a proliferation and ER α western in CAMA-1 cells. Since the results of the proliferation of Her2/neu cells and ZR-75-1 cells did not include the other analogs, these were removed. We have also reorganized the figures, moving the xenograft data to after the structural analysis. This new figure highlights the clinical potential of OP-1074 (only) and includes our Y537S mutant data, which many resistant tumors contain.

- Reviewer 2 stated: “The authors performed tamoxifen resistant MCF7 xenograft experiment to evaluate the efficacy of anti-cancer effect of OP-1074 in vivo compared to tamoxifen and fulvestrant. They concluded that the OP-1074 repressed the growth of tamoxifen resistant MCF7 cells more effectively than tamoxifen or fulvestrant (Fig.3). However, they treated OP-1074 twice daily in spite of others were once daily. This experimental design is quite questionable to describe their conclusion. To prevent misunderstanding of readers, the authors should emphasize this different treatment scheme. Or the authors should reevaluate the results using the same treatment scheme. Moreover, the authors need to show experimental data of pharmacokinetic analysis (p.10).”
 - The paragraph was revised and a sentence was added to emphasize and justify dosing: “In the absence of prior pharmacokinetic data for OP-1074, animals were treated with 100 mg/kg twice a day by oral gavage, whereas fulvestrant was treated once a day subcutaneously at the same dose, and this extra daily dose of OP-1074 may explain some of the enhanced efficacy observed after day 27 of this study”. Additionally, a graph of the experimental data of the pharmacokinetic analysis was added to the final figure.
- Reviewer 2 stated: “To verify the chemical dependent H12 movement, the authors employed Trypsin-coupled fluorescent polarization assay. The experimental results are shown in supplemental file S4. This result is important for suggesting that OP-1074 enhanced H12 movement. However, the result is unclear to convince this reviewer. If OP-1074 has similar activity to fulvestrant, profile should be similar to fulvestrant (ICI) or Apo in fig.S4. However, the profile of OP-1074 is similar to other OP chemicals instead of fulvestrant or Apo. It is hard to understand why the X axis of fig S4 using the log₂ scale. It is just 3, 4, 5, 6, 7, 8 min. How did the authors evaluate Apo and fulvestrant mediated half-life (2.2 min and 2.3 min respectively) from this experiment? They have to show us understandable results.
 - We have reworded parts of this section to emphasize the fact that OP-1074 does not disrupt H12 to the extent of fulvestrant. Rather, the disruption of H12 by OP-1074 appears sufficient to elicit ER α degradation. We have also reworked the figures to make them more legible. We also included a sentence describing how the half-life is calculated.
 - The log₂ scale was used to highlight the differences in the slopes of the fits between the curves. It is the slope that gives the half-life number.
- Reviewer 2 stated: The authors analyzed agonist and antagonist activity of OP-1074 and described in the results “OP-1074 lacked activity on other nuclear receptors and...”. This sentence should be “OP-1074 lacked activity on other steroid hormone receptors and...” (p.6).
 - This has been fixed.
- Reviewer 2 stated: The authors show IC₅₀ in the figures as log₁₀ (there is no description). It is unkind for the readers. It should be described as nM (Fig. 2 and Fig.s2).
 - Data has been converted into nM to help with consistency and readability.
- Reviewer 2 stated: There is no endoxifene (Endox) result in Fig2B and FigS3 despite it is described in the text. The authors have to show the Endox results to compare the results of Fig.1.
 - Mention of endoxifen has been removed from the text describing Figure 2, and Figure S3 has been removed. As a reference SERM it remains in the master table, Table 1, but was removed from the text because its relative potency to the analogs was not germane. The text has been re-written to highlight the similarities of the potencies of the analogs to each other, revealing that the position of the methyl group does not affect antagonism in the breast, an AF-1 independent tissue. We contrast this with the differences in activities on AP, an AF-2 dependent gene.

- Reviewer 2 stated: There is no information of treated chemical concentration for ER α western blot analyses (Fig2E, FileS3C, D). The authors should describe information. There is unrelated experiment statement in supplement figure S3 legend.
 - o Doses of ligands that were used to treat cells loaded in the immunoblots were added to figure legends (they were already in the methods).
- Reviewer 2 stated: The label of the X axis in Fig4A is not proper. It should be in log(nM).
 - o This has been fixed.
- Reviewer 3 stated: “There are several instances of overreaching or incorrect statements. In the paper that the authors cite for the ability of the AP assay to discriminate between pure antiestrogens and SERMs, this assay classified the bioactive tamoxifen metabolite monohydroxytamoxifen as pure antagonist, even so tamoxifen functions as a SERM.”
 - o The language has been improved to make the differences between the two clearer.
- Reviewer 3 stated: Similarly, endoxifen is not “the” bioactive metabolite of tamoxifen, but rather “one” of the metabolites, and the authors switch between endoxifen in the AP assay and 4-hydroxytamoxifen (4-OHT) in other assays. Ideally, the authors should include both metabolites in all in vitro assays. Similarly, the statement that “OP-1074 was slightly less potent than Fulvestrant, and more potent than 4-OHT” is biased as the potency was exactly in the middle between that of Fulvestrant and 4-OHT.
 - o Comparisons that may be construed as biased have been fixed and/or removed. The description of endoxifen was changed to the “*most abundant* bioactive metabolite of tamoxifen” and references were added to support this claim. With the changes discussed above there will not be any mention of tamoxifen in that section to contend with.
- Reviewer 3 stated: The authors should clarify that pure antiestrogens are not always advantageous over SERMs. Rather, tamoxifen has remained the first line treatment of ER-positive breast cancer, its agonist activity in bone is clearly beneficial, and endoxifen has been found to be effective in patients who have failed Fulvestrant treatment.
 - o A sentence has been added in the conclusion discussing possible effects of OP-1074 on bone, a tissue in which SERMs are at an advantage. A sentence was added to the introduction discussing the results from recent phase I study in which endoxifen showed clinical benefit in patients with prior fulvestrant treatment.
- Reviewer 3 stated: Fig. S3B: Please show actual data, rather than just a summary table.
 - o We believe that the reviewer was referring to supplemental figure 2, panel B. We have fixed the table for Supplement S2b to show actual data and replicates.
- Reviewer 3 stated: Please label the main helices and loops in Figs. 5 and 6.
 - o We labeled the main helices.
- Reviewer 3 stated: Given the proposed critical role of the stereospecific methylpyrrolidine group, the authors should prepare a figure to show the interaction of the methyl group with the receptor residues. Also, a figure for the comparison of OP-1074 complex with that of tamoxifen/hydroxyl tamoxifen will be helpful.
 - o New panels have been added to figure 5 that highlight the interaction between the pyrrolidine with the receptor residues for each ligand.
 - o A new supplemental figure, Supp. File Figure 4, has been added that shows an overlay of the OP-1074 x-ray crystal structure with an endoxifen-bound structure (PDB: 5W9D). Comparison between these structures is more appropriate than with the 4-hydroxytamoxifen structure (PDB: 3ERT) because it uses the same protein construct and crystallographic space group.
- Reviewer 3 stated: In panel D of Figure 5, the distance of 1.8 Å for a hydrogen bond is too short.
 - o This issue has been fixed and the distance is now correct for a hydrogen bond at ~ 3Å.

- Reviewer 3 stated: Suppl File S5: the authors should include the high resolution shell data of Rmerge, completeness, and redundancy. Again, the R factors of OP-1156 complex are a little high based on the 2 Å resolution of the data.
 - o These data have been included in the table. We were able to further refine the structure and get the Rfree from ~28 to 26.84. We were unable to improve the R-free further. We believe that this is because there is a loop around residue 333 and another loop around residue 460 that is disordered.
 - o The new coordinates have been uploaded to PDB and will supersede the old ones for OP-1156.

Thank you very much for your consideration.

Reviewers' Comments:

Reviewer #1 (Remarks to the Author):

The authors synthesized five benzopyrans with variable side chains to identify pure antiestrogens in uterotrophic assay. Compound OP-1074 was identified as a pure antiestrogen and selective ER degrader. Oral administration of OP-1074 proved to be more efficacious than fulvestrant in shrinking tumors in a tamoxifen resistant xenograft model. Further biochemical and crystal structure analyses revealed a structure activity relationship implicating the importance of a stereospecific methyl on the pyrrolidine side chain of OP-1074. The work is indeed of interest, and systematical evaluations were conducted, revealing that compound OP-1074 does exhibit better properties than drug fulvestrant. Although only methyl substituted pyrrolidines were studied, the identification of OP-1074 and the stereochemistry effects on optimal orientation of ligand binding to ER α to induce complete antiestrogenicity is interesting, which would provide guidance for future drug design for treating endocrine-resistant tumors. Further optimization of the substituent effects of pyrrolidines would be more interesting. This reference recommend the work to be accepted for publication.

Reviewer #2 (Remarks to the Author):

This revised manuscript is improved and understandable. This reviewer agrees with the anti-estrogenic effect of OP-1074 and it causes stimulation of ER α proteolysis.

However, there remains the frustration of insufficiency of the mechanism(s) of ER α proteolysis. The differential results of a trypsin-coupled fluorescence polarization assay and a crystallographic analysis between fulvestrant and OP-1074 suggest that the mechanism of ER α proteolysis and anti-estrogenic activity of those chemicals might be different. It has been reported that fulvestrant enhances ER α proteolysis, at the same time fulvestrant affects the nuclear-cytoplasmic distribution of ER α (Mahoudi et al., PNAS 1995, Long and Nephew JBC 2006). These features cause a strong anti-estrogenic activity of fulvestrant different from tamoxifen. This report shows the influence of the stereochemical factor on the helix 12 flexibility/dislocation, however there is no information regarding the connection between OP chemical mediated helix 12 dislocation and ER α proteolysis mechanism. It is a weakness of this report.

The reference for the structure of rat ER beta with ICI should be [26, Pike, A.C.W 2001] not [25].

Reviewer #3 (Remarks to the Author):

Overall, the authors have not been very responsive to this reviewer's (#3) comments, e.g.:

1. In response to Reviewer #2, the authors still claim that the AP assay "robustly predicts" compounds as either SERMs or full antagonists, for which they cite 2 papers. I don't understand how the authors can make such a strong claim, when the paper they cite (Holinka et al.) predicts the SERM 4-OHT as full antagonist based on this assay.

2. In response to this reviewer's request to show the actual data on which the table in fig. S2 is based, they show an updated table with EC50 values and unidentified error (SD or SEM)? Please show the actual data, which are the full dose-response curves.

3. Labeling of the structure figures is still poor and H12 residues that interact with the critical, stereo-specific methyl group are still not indicated.

Response to Reviewers

We have addressed these comments and have made the following changes to the previously submitted manuscript:

Reviewer #1 (Remarks to the Author):

The authors synthesized five benzopyrans with variable side chains to identify pure antiestrogens in uterotrophic assay. Compound OP-1074 was identified as a pure antiestrogen and selective ER degrader. Oral administration of OP-1074 proved to be more efficacious than fulvestrant in shrinking tumors in a tamoxifen resistant xenograft model. Further biochemical and crystal structure analyses revealed a structure activity relationship implicating the importance of a stereospecific methyl on the pyrrolidine side chain of OP-1074. The work is indeed of interest, and systematical evaluations were conducted, revealing that compound OP-1074 does exhibit better properties than drug fulvestrant. Although only methyl substituted pyrrolidines were studied, the identification of OP-1074 and the stereochemistry effects on optimal orientation of ligand binding to ER α OP-1074 does exhibit antiestrogenicity is interesting, which would provide guidance for future drug design for treating endocrine-resistant tumors. Further optimization of the substituent effects of pyrrolidines would be more interesting. This reference recommend the work to be accepted for publication.

- We would like to thank this Referee for their constructive comments.

Reviewer #2 (Remarks to the Author):

This revised manuscript is improved and understandable. This reviewer agrees with the anti-estrogenic effect of OP-1074 and it causes stimulation of ER α proteolysis.

However, there remains the frustration of insufficiency of the mechanism(s) of ER α the anti-estrogenic effect of OP-1074 and it rypsin-coupled fluorescence polarization assay and a crystallographic analysis between fulvestrant and OP-1074 suggest that the mechanism of ER α of the mechanism(s) of ER α the anti-estrogenic effect of OP-1074 and it rypsin-coupled fluorescencifulvestrant enhances ER suggest that the mechanism of Efulvestrant affects the nuclear-cytoplasmic distribution of ER α (Mahoudi et al., PNAS 1995, Long and Nephew JBC 2006). These features cause a strong anti-estrogenic activity of fulvestrant different from tamoxifen. This report shows the influence of the stereochemical factor on the helix 12 flexibility/dislocation, however there is no information regarding the connection between OP chemical mediated helix 12 dislocation and ER α helix 12 dislocation weakness of this report.

- We thank this Referee for their constructive comments. We agree that the mechanism of proteolysis elicited by OP-1074 (and other oral SERDs) is important but we believed that it was outside of the scope of this work. It is research that we will pursue and believe that

it will answer important questions about how oral SERDs achieve activity in breast cancer.

The reference for the structure of rat ER beta with ICI should be [26, Pike, A.C.W 2001] not [25].

- This reference has been fixed.

Reviewer #3 (Remarks to the Author):

Overall, the authors have not been very responsive to this reviewer's (#3) comments, e.g.:

1. In response to Reviewer #2, the authors still claim that the AP assay "robustly predicts" compounds as either SERMs or full antagonists, for which they cite 2 papers. I don't understand how the authors can make such a strong claim, when the paper they cite (Holinka et al.) predicts the SERM 4-OHT as full antagonist based on this assay.

- We thank the reviewer for pointing out any discrepancies in the literature and have added new references to support our claim. The sentence in which Holinka paper is referenced was intended to support the claim that "the AP assay correlates with ER-mediated increase in uterine wet weight in vivo", which it does by showing that several types of estrogens robustly activates AP over a large range of concentrations over several days. They are also the group that originally developed the assay so we believe they deserve credit for this claim. We agree that it is a weakness in the reference that 4-OHT only activated AP a small amount, only 1.3-fold. To compensate for this weakness we have added 2 more references that clearly demonstrate that the AP activity of 4-OHT and Fulvestrant correlates with uterine wet weight in vivo.

2. In response to this reviewer's request to show the actual data on which the table in fig. S2 is based, they show an updated table with EC50 values and unidentified error (SD or SEM)? Please show the actual data, which are the full dose-response curves.

- We have added the dose response curves to this figure.

3. Labeling of the structure figures is still poor and H12 residues that interact with the critical, stereo-specific methyl group are still not indicated.

- We have added more amino acid/helix labels to the structure figures and have made them larger to aid in interpretation. We have also changed the wording in the second paragraph of the "3R-Methylpyrrolidine disrupts helix 12" section of the results to "...which propagates to helix 12" to clarify our analysis of the structure.